# Bioinspired nacre-like alumina with a bulk-metallic glass-forming alloy as a compliant phase

Amy Wat[1,2], Je In Lee[3,4], Chae Woo Ryu[3], Bernd Gludovatz[5], Jinyeon Kim[3,6], Antoni P. Tomsia[2], Takehiko Ishikawa[7], Julianna Schmitz[8], Andreas Meyer[8], Markus Alfreider[9], Daniel Kiener [9], Eun Soo Park [3] & Robert O. Ritchie [1,2]

Bioinspired ceramics with micron-scale ceramic "bricks" bonded by a metallic "mortar" are projected to result in higher strength and toughness ceramics, but their processing is challenging as metals do not typically wet ceramics. To resolve this issue, we made alumina structures using rapid pressureless infiltration of a zirconium-based bulk-metallic glass mortar that reactively wets the surface of freeze-cast alumina preforms. The mechanical properties of the resulting $Al_2O_3$ with a glass-forming compliant-phase change with infiltration temperature and ceramic content, leading to a trade-off between flexural strength (varying from 89 to 800 MPa) and fracture toughness (varying from 4 to more than 9 MPa·m$^{½}$). The high toughness levels are attributed to brick pull-out and crack deflection along the ceramic/metal interfaces. Since these mechanisms are enabled by interfacial failure rather than failure within the metallic mortar, the potential for optimizing these bioinspired materials for damage tolerance has still not been fully realized.

[1] Department of Materials Science and Engineering, University of California, Berkeley, CA 94720, USA. [2] Materials Sciences Division, Lawrence Berkeley National Laboratory, Berkeley, CA 94720, USA. [3] Research Institute of Advanced Materials, Department of Materials Science and Engineering, Seoul National University, Seoul 08826, Republic of Korea. [4] International Center for Young Scientists, National Institute for Materials Science, 1-2-1 Sengen, Tsukuba, Ibaraki 305-0047, Japan. [5] School of Mechanical and Manufacturing Engineering, UNSW Sydney, Sydney, NSW 2052, Australia. [6] Advanced Analysis Center, Korea Institute of Science and Technology, Seoul 02455, Republic of Korea. [7] Japan Aerospace Explanation Agency, 2-1-1 Sengen, Tsukuba, Ibaraki 305-8505, Japan. [8] Institut für Materialphysik im Weltraum, DLR, Köln 51170, Germany. [9] Department of Materials Science, Montanuniversität Leoben, Leoben 8700, Austria. These authors contributed equally: Amy Wat, Je In Lee. Correspondence and requests for materials should be addressed to E.S.P. (email: espark@snu.ac.kr) or to R.O.R. (email: roritchie@lbl.gov)

New materials for structural applications in aerospace, energy, and transportation often have the requirement to operate safely at high temperatures in aggressive environments and, for aviation applications, have low density. Ceramic materials in many respects represent an ideal solution to this problem, but their use has been severely compromised by the fact that they invariably display near-zero tensile ductility and low fracture toughness values, which makes them prone to sudden catastrophic failure.

Nature, however, is particularly adept at designing damage-tolerant ceramic-like materials with excellent strength and toughness, using a small palette of individual constituents with relatively meager mechanical properties. Nature develops remarkable materials through sophisticated hierarchical, multiple length-scale architectures that optimize the mechanical properties of the hard mineral and soft organic phases, often with compositional, orientation, or structural gradients and graded interfaces[1–4]. A notable example here is nacre, which is known to have a fracture toughness three orders of magnitude higher (in energy terms) than its constituents. Nacre is found in mollusk shells, such as abalone, sea snails, and various bivalves. The particularly well-studied variety found in abalone shells is comprised of ~95 vol.% aragonite mineral (calcium carbonate) and ~5 vol.% biopolymer[1,2,4,5], formulated into a brick-and-mortar microstructure that enables multiple toughening and strengthening mechanisms.

The brick-and-mortar microstructure facilitates the creation of damage tolerance. The mineral "bricks" provide strength, whereas limited (a few micrometer) displacements within the biopolymeric "mortar" act to dissipate locally high stresses, thereby providing a degree of ductility that promotes toughness. The salient toughening mechanisms are principally extrinsic[6] and involve crack deflection and primarily brick pullout leading to crack bridging[4]. The sliding behavior within the mortar is essential for toughening, but it must be limited to retain strength[1]. It is restricted due to the roughness on the platelet surfaces and the tensile and shear strength of the biopolymers that act as a glue between the platelets[7]; in certain organisms, tablet interlocking due to the dovetail geometry[8] of the platelets and mineral bridges that connect the platelets[4,9] can also inhibit platelet sliding. Nature's precise tailoring of its structures and constituents' properties, e.g., the use of high-aspect mineral bricks with a tensile/shear-resistant mortar can generate mechanical performance that is comparable with advanced engineering ceramics[5]. Computational models and biomimetic approaches therefore suggest that there is potential to create new lightweight structural materials with current engineering ceramics by utilizing the design principles and strength/toughening mechanisms active in nacre and other biological materials[8–10].

A promising method to recreate nacre-like materials using ceramics is freeze-casting followed by infiltration of a compliant (mortar) phase, such as a polymer or metal[11–13]. Freeze-casting is a method of creating a porous ceramic scaffold by mixing ceramic particles in water and then freezing the water with a temperature differential to induce lamellar ice growth. Once the ice freezes, it can be sublimated leaving a porous ceramic structure in the negative image of the ice structure[14]. This technique can create ceramic materials with complex hierarchical microstructures that can be controlled using various casting conditions and additives in the solvent[15–20]. One notable example is a freeze-cast alumina scaffold that was pressed and infiltrated with polymethyl methacrylate as a mortar to mimic the nacre architecture[11]. With 80 vol.% ceramic in a brick-and-mortar structure, it displayed a strength of ~225 MPa and an extremely high toughness in excess of 30 MPa·m$^{1/2}$, making it one of the toughest ceramic materials reported to date. Indeed, there have been comprehensive review papers on various experiments to freeze-cast[21] or to use other methods[22,23] to create such nacre-like ceramics.

There has been sustained interest in creating ceramic structures with a nacreous microstructure, particularly as computational models have suggested that the strength of nacre-inspired ceramics could increase, without sacrificing toughness, if the polymer-compliant phase was replaced by a metal[8]. This direction has the best potential to create exciting new lightweight structural materials with high strength and toughness, and a capability for high-temperature functionality. To make such biomimetic materials, methods such as coextrusion[24–26], spark-plasma sintering[27], and additive manufacturing[28] have been attempted with varying success due to issues with undesirable reactions, dewetting between the metal and the ceramic phases, or coarseness of the ceramic brick phases. Melt-infiltration after freeze-casting a ceramic scaffold represents a liquid-processing technique that has the potential to fabricate ceramic–metal hybrid materials with strong interfacial bonding[29]. Hybrid ceramics with a lamellar structure have been fabricated by pressure-assisted infiltration with conventional Al alloys[30–32], but those contained less than 40 vol.% of ceramic, which is significantly lower than the mineral content in nacre. Furthermore, as the volume fraction of reinforcement increases and the pore spacing decreases in a scaffold, the pressure required to completely infiltrate the scaffold increases, leading to deformation or cracking of the scaffold[33,34]. Since the magnitude of the applied pressure is closely related to the wettability of the ceramic by a molten alloy, strategies such as the addition of a reactive element in the matrix alloy or coating the surface of reinforcements have been applied to improve the wettability and therefore feasibility of infiltration between the alloy and ceramic[33]. For example, Mg was used as a reactive element to fabricate bioinspired hybrid materials using pressureless infiltration[28,35–37], but the microstructure was lamellar with a low ceramic content compared to the brick-and-mortar structure.

In an attempt to solve this problem, we use reactive wetting to create a nacre-like, high volume-fraction ceramic material containing a metallic-compliant phase. We selected a Zr-based bulk-metallic glass (BMG) as the mortar phase, as this alloy shows excellent wettability with alumina[38,39]. BMGs are metallic alloys with a disordered, noncrystalline atomic structure that can be cast in bulk (over 1 mm) layers. Zr-based BMGs have been shown to have high strength with some degree of toughness, which represent ideal properties as a compliant-phase (mortar) material. As a result, the combination of perfect wettability and mechanical properties of the BMG makes it an interesting candidate for infiltration into a brick-like ceramic scaffold. However, processing of BMGs introduces challenges as they can oxidize at elevated temperatures[40] and require a critical cooling rate to solidify in the amorphous state. Moreover, the final material must be used below the glass-transition temperature as these glasses can embrittle upon crystalization[41]. While fibrous or particulate BMG-matrix composites have been developed by melt-infiltration[42,43], this study relates their infiltration behavior to the thermophysical properties and wettability of the BMG, in order to tailor the properties of the ceramic/metal interfaces while maintaining the BMG in the fully amorphous state as a metallic mortar in nacre-like material.

Here, we synthesize "nacre-like" (high-volume fraction) alumina structures, containing a glass-forming alloy as the compliant phase, processed via pressureless infiltration. Contact-angle measurements and electrostatic levitation were used to identify the length scale of the ceramic scaffolds and their excellent wettability with a liquid alloy as crucial factors to successfully synthesize these materials. Near-perfect wetting within seconds for the Zr-based BMG on alumina was observed, indicating that a

conformal bond can be attained at the interface of the two materials, which in turn implies that high strength may be realized in an $Al_2O_3$/BMG hybrid material[44,45]. However, the interfacial strength can markedly change due to the formation of brittle interfaces upon solidification of the metallic phase, which can compromise the damage tolerance of the bioinspired material. Finally, we examine how processing conditions can affect the ceramic–metal interfaces to discern the fundamental mechanisms underlying the flexural strength and fracture toughness of these bioinspired nacre-like ceramics.

## Results

**Thermophysical and wetting properties.** Surface tension ($\gamma$) and viscosity ($\eta$) of the $Zr_{46}Cu_{30.14}Ag_{8.36}Al_8Be_{7.5}$ BMG-forming alloy were evaluated by electrostatic levitation (ESL) over temperatures between 1180 and 1310 K. The surface tension $\gamma$ of the melt showed a negative temperature dependence (Fig. 1a) that can be described, in terms of the absolute temperature $T$ (in kelvin), as

$$\gamma = 1.69 - 4.07 \times 10^{-4} T, \tag{1}$$

and a negative temperature dependence of the viscosity $\eta$ (Fig. 1b), which can be described by the Vogel–Fulcher–Tammann relationship[46] as

$$\eta = \eta_0 \exp\left(\frac{D^* \cdot T_0}{T - T_0}\right), \tag{2}$$

where $\eta_0$ is the viscosity in the infinite temperature limit ($2.03 \times 10^{-4}$ Pa·s), $D^*$ is the fragility parameter (5.87), and $T_0$ is the absolute temperature (598 K) where the viscosity becomes infinite[46]. The wettability of alumina by the BMG was evaluated by the sessile-drop method at 1153 and 1273 K (Fig. 2). As shown in the inset of Fig. 2, the alloy melt that was dropped on the alumina substrate exhibited initial contact angles of ~14°, and decreased continuously within several tens of seconds. The final contact angles were 8° at 1153 K and 6° at 1273 K, showing the excellent wetting behavior of alumina by the BMG-alloy melt.

**Microstructure.** Figure 3a shows the phase constitution of the compliant-phase alumina/BMG materials with a lamellar structure infiltrated at 1153 and 1273 K. In both curves, the peaks of crystalline phases were superimposed on the broad diffuse scattering, indicating the presence of large fractions of the amorphous phase in the metallic mortar after the melt-infiltration followed by water quenching. Both materials had dominant peaks of alumina and $Zr_2Cu$, which is one of the intermetallic compounds appearing in the $Zr_{46}Cu_{30.14}Ag_{8.36}Al_8Be_{7.5}$ BMG-forming alloy as heterogeneous nucleation occurs during solidification[47,48]. The peak intensity of the $Zr_2Cu$ phase increased as the infiltration temperature increased. Peaks of $ZrO_2$, which is one of the reaction products between alumina and the Zr-based BMG-forming alloy melt[39], were observed in the sample infiltrated at 1273 K.

The corresponding alumina/BMG materials with a brick-and-mortar structure (Fig. 3b) had a similar phase constitution as the lamellar samples, but showed a reduced peak intensity of the metallic phases due to the decreased volume fraction of the metallic mortar. These results suggest that the phase constitution of these alumina/glass-forming alloy hybrid materials is independent of the alumina volume fraction, but dependent on the infiltration temperature.

Figure 4a, c shows the microstructure of the alumina-based hybrids with a lamellar structure. A lamellae thickness of $7.7 \pm 4.1\,\mu m$, metal thickness of $13.3 \pm 4.7\,\mu m$, and alumina content of 37 vol.% were measured using image analysis. Figure 4b, d shows

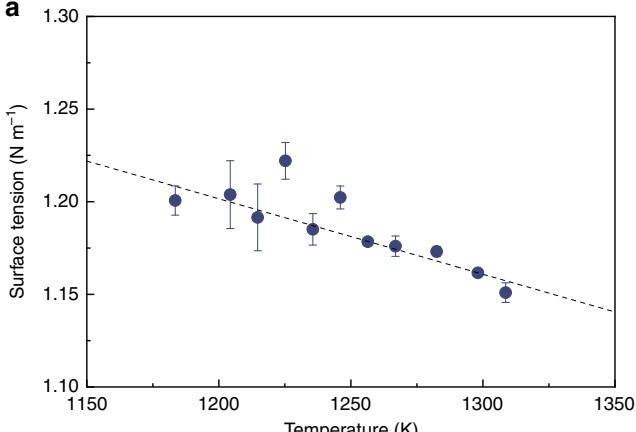

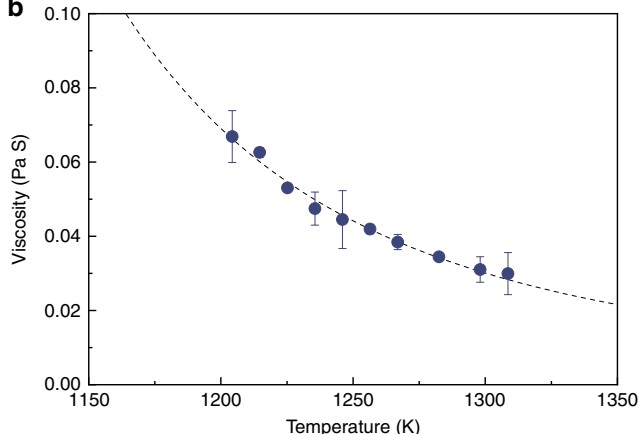

**Fig. 1** Thermophysical properties of the bulk-metallic glass. **a** Surface tension ($\gamma$) and **b** viscosity ($\eta$) of the $Zr_{46}Cu_{30.14}Ag_{8.36}Al_8Be_{7.5}$ BMG-forming alloy melt measured by electrostatic levitation (ESL) over the temperature range from 1180 to 1310 K. The surface tension decreases by a mere 5% over the temperature range, while the viscosity for the melt decreases by ~50% with increasing temperature

the corresponding microstructure of the hybrids with brick-and-mortar structures, formed after pressing the alumina with paraffin wax and subsequently infiltrating with the $Zr_{46}Cu_{30.14}Ag_{8.36}Al_8Be_{7.5}$ BMG as the compliant (mortar) phase. The structures exhibit a brick thickness of $22.6 \pm 11.2\,\mu m$, a metallic mortar thickness of $4.4 \pm 3.4\,\mu m$, and an alumina content of 80 vol.%. All the materials prepared by pressureless infiltration have a well-formed lamellar and brick-and-mortar microstructure, apparently with complete infiltration, without pores, of the metallic mortar at both infiltration temperatures. Akin to natural nacre, both the lamellar and brick-and-mortar microstructures display pre-existing bridges that act as local connections between the alumina lamellae or between the bricks. The BMG–ceramic interfaces in the materials infiltrated at 1153 and 1273 K are imaged in Fig. 4e–h. The materials infiltrated at 1153 K (Fig. 4e, g) show sharp interfaces between the alumina (black) and the amorphous phase (gray), whereas the materials infiltrated at 1273 K (Fig. 4f, h) display evidence of erosion of the alumina surface with voids at the interface. The material with a lamellar structure infiltrated at 1273 K (Fig. 4f) contains a faceted crystalline phase (dark gray) grown from the interface into the amorphous phase. This crystalline phase was identified as the intermetallic compound $Zr_2Cu$ by energy-dispersive X-ray spectroscopy (EDS) analysis, as shown in the X-ray diffraction (XRD) analysis in Fig. 3. The material with brick-and-mortar

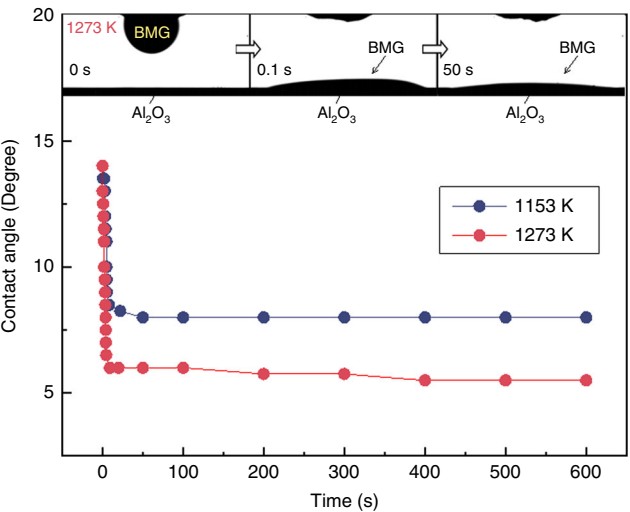

**Fig. 2** Rate and extent of wetting of bulk-metallic glass on alumina. Wetting angles, measured using a sessile-drop experiment, at temperatures of 1153 and 1273 K. The wetting angles, when plotted with respect to time, show that the contact angle converges on 8° at 1153 K and 6° at 1273 K within 10 s. This is a strong indication that the wetting kinetics are independent of any potential reaction occurring at the interface between the metal melt and the ceramic solid

structure infiltrated at 1273 K showed an irregular grain morphology in the metallic mortar, indicating the formation of a multiphase microstructure with $Zr_2Cu$ and unknown phases, as shown in the XRD analysis. These results imply that the interfacial structure of our materials significantly changes as the infiltration temperature increases.

Differential scanning calorimetry (DSC) traces of the alumina/glass-forming alloy hybrids with different structures and infiltration temperatures are given in Fig. 5. All materials showed a decrease in the crystallization onset temperature ($T_x$) compared with the monolithic BMG. The reduced thermal stability of the amorphous phase is attributed to the presence of the BMG–ceramic and BMG–intermetallic compound interfaces that act as heterogeneous nucleation sites during heating[47]. All materials exhibited a similar glass-transition temperature, $T_g$, of 705 K and crystallization onset temperature, $T_{x.Hybrid}$, of 790 K. This indicates that the composition of the amorphous phase in the metallic mortar is very similar to that in the alumina/glass (hybrid) materials. The heat of crystallization was measured as 74.4 J·g$^{-1}$ and 41.5 J·g$^{-1}$ for lamellar structures, and 52.5 J·g$^{-1}$ and 2.2 J·g$^{-1}$ for brick-and-mortar structures, respectively, infiltrated at 1153 K and 1273 K; all these values are lower than for monolithic BMG. The results indicate that the volume fraction of the amorphous phase in the metallic mortar is lower, while that of the crystalline phase increases with increasing infiltration temperature.

**Mechanical properties**. The flexural strength of the alumina hybrid materials was found to be markedly affected by the infiltration temperature used to produce the material. Lamellar samples infiltrated at 1153 K had a flexural strength of ~797 ± 60 MPa, whereas those infiltrated at 1273 K displayed a dramatically lower value of ~225 ± 11 MPa. The brick-and-mortar specimens had a flexural strength of ~366 ± 168 MPa when infiltrated at 1153 K and again showed a much lower strength, of ~88 ± 28 MPa, when infiltrated at 1273 K. For both brick-and-mortar and lamellar structures, infiltrating at a higher temperature resulted in an approximate fourfold decrease in strength,

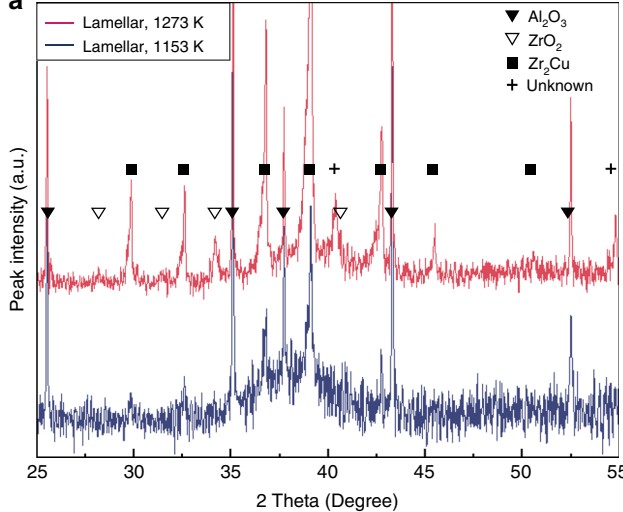

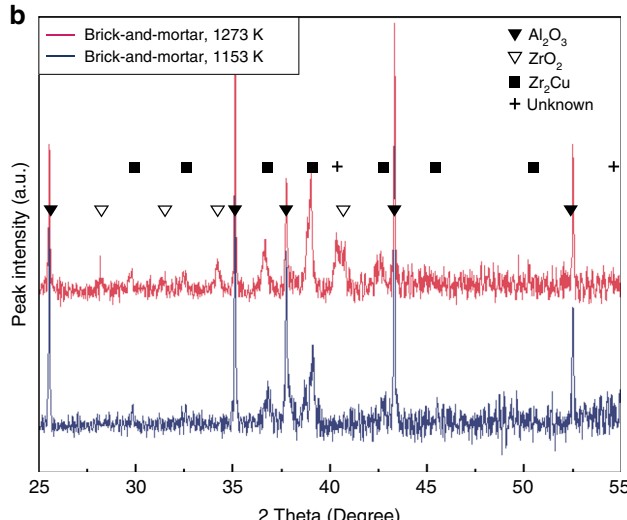

**Fig. 3** X-ray diffraction (XRD) of compliant-phase ceramics. **a**, **b** XRD patterns of the alumina/BMG compliant-phase ceramics, with a lamellar structure (**a**) and brick-and-mortar structure (**b**), infiltrated at 1153 and 1273 K. In both curves, the dominant peaks of alumina and $Zr_2Cu$ were superimposed on the broad diffuse scattering, indicating the presence of an amorphous phase in the metallic mortar after the melt infiltration followed by water quenching. The results indicate that the phase constitution of the alumina/BMG-forming materials is dependent on the infiltration temperature, but independent of the alumina volume fraction

compared with those infiltrated at 1153 K (Fig. 6a). The strength was also affected by the ceramic content of the materials; alumina has a flexural strength of ~330 MPa, compared with the BMG at 1700 MPa. Accordingly, the strength of the brick-and-mortar samples, with their far higher volume fraction of ceramic, was significantly lower than that of the lamellar samples. This is very pronounced in the stress–strain curves, as shown in Supplementary Fig. 1a.

Conversely, there was little change in the fracture toughness of the lamellar samples infiltrated at 1153 K and at 1273 K, which were measured to be 5.7 ± 0.9 MPa·m$^{1/2}$ and 4.2 ± 0.9 MPa·m$^{1/2}$, respectively (Fig. 6b); indeed, they are comparable when considering their standard deviations. However, the brick-and-mortar samples show very different properties. Brick-and-mortar samples infiltrated at 1153 K failed catastrophically with a

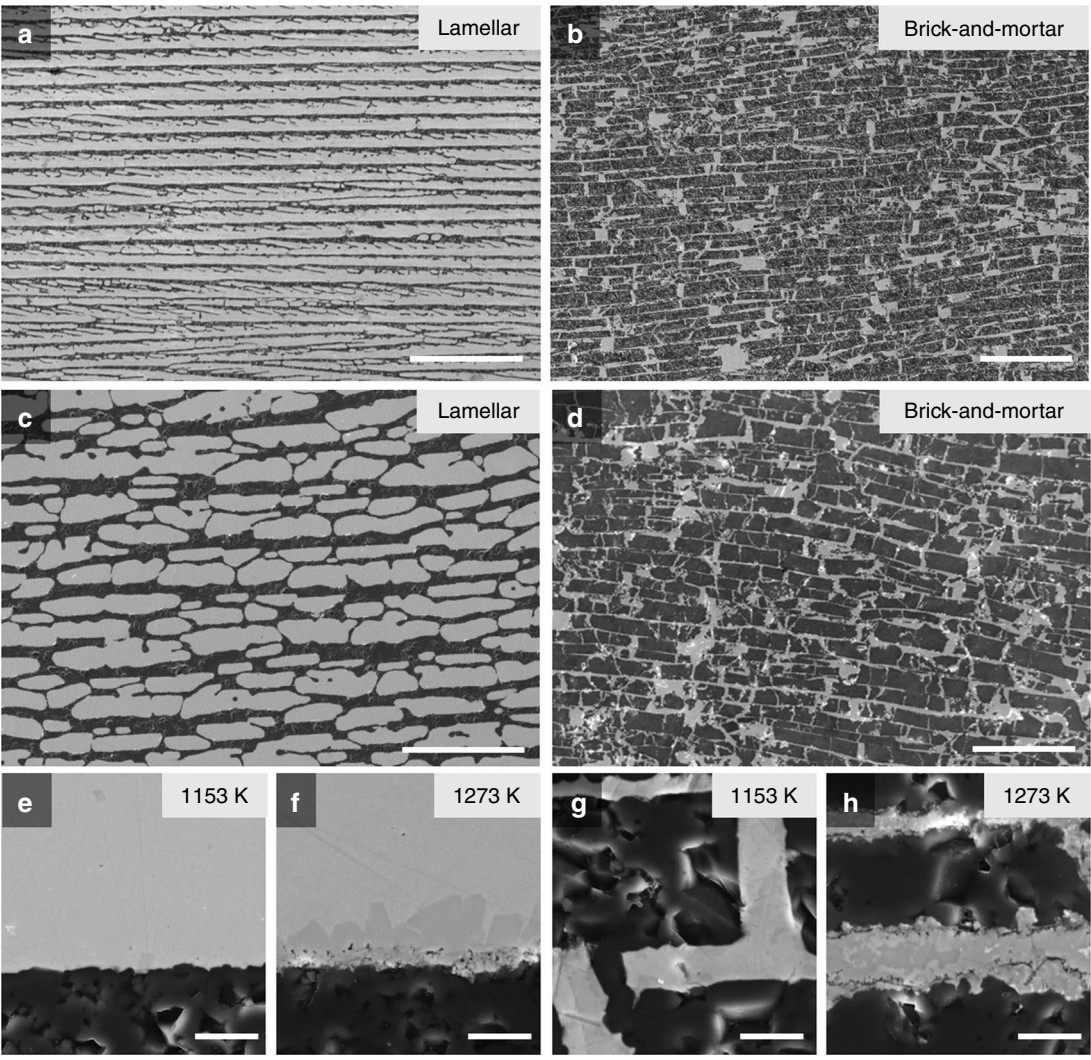

**Fig. 4** SEM micrographs of the alumina/BMG-alloy materials. **a–d** Lamellar (**a**, **c**) and brick-and-mortar structures (**b**, **d**). **e–h** The metal–ceramic interfaces in these ceramics infiltrated at 1153 K and 1273 K are shown in **e** and **f** for a lamellar structure and **g** and **h** for brick-and-mortar structures, respectively. The materials infiltrated at 1153 K (**e**, **g**) show a sharp interface between the alumina (black) and amorphous phase (gray), but the materials infiltrated at 1273 K (**f**, **h**) show erosion of the alumina surface and voids at the interface. Scale bars: 300 μm (**a**, **b**), 100 μm (**c**, **d**), 5 μm (**e**, **f**, **g**, **h**)

fracture toughness of ~5 MPa·m^½ with no stable crack growth. In contrast, those infiltrated at 1273 K displayed stable crack growth with crack extensions of almost 1 mm along the interface between the ceramic and metallic phases. As a result, the fracture toughness of these brick-and-mortar materials displayed markedly rising R-curve behavior (Fig. 7), with the toughness increasing from the intersect of the blunting line of roughly 5–8 MPa·m^½ after ~200 μm of stable crack growth and ultimately reaching as high as 14 MPa·m^½ (although, as Fig. 7 indicates, the higher toughness number is not strictly valid in terms of ASTM Standard E1820[49]). Representative load–displacement curves for these tests can be found in Supplementary Fig. 1b, c. Scanning electron microscope (SEM) micrographs in Fig. 8b, d reveal that both lamellar and brick-and-mortar samples infiltrated at 1273 K failed with multiple crack-path deviations off the plane of maximum tensile stress caused by crack deflections along the ceramic/metal interfaces; this creates a rough fracture surface with significant evidence of pullout of the ceramic bricks, and consequently higher toughness. In contrast, all samples infiltrated at 1153 K in Fig. 8a, c failed catastrophically directly at the onset of crack initiation.

To address the individual interfaces in more detail, micro-cantilever experiments, as depicted in Fig. 9a, were conducted on lamellar as well as brick-and-mortar samples of both infiltration temperatures. Notably, whereas the hierarchical structure exhibited no influence on the local interfacial fracture behavior, the change in infiltration temperature resulted in a major discrepancy. Figure 9b shows the conditional stress intensity–displacement data for 1153 and 1273 K specimens. Due to the nonlinear elastic behavior and the fact that the 1153 K sample did not break at the interface, these values are to be taken predominantly as a means of comparison between each other. Nevertheless, the lower stress intensity at failure of the 1273 K specimen goes hand in hand with the fact that the adhesion between the $Al_2O_3$ matrix and the BMG mortar was rather weak. The interface fracture surfaces (Fig. 9d, f) show high fractions of asperities and individual particles not connected to the BMG mortar. Conversely, the 1153 K specimen did not break at the interface but the crack extended through the weaker BMG in a stable manner, leaving behind a large plastically deformed zone, as depicted in Fig. 9c, e.

## Discussion

In this work, we have created a freeze-cast alumina ceramic with a bioinspired, nacre-like, brick-and-mortar structure with a Zr-based BMG as the infiltrated compliant-phase mortar. Our motivation is that nacre-like alumina infiltrated with a polymer-compliant phase has been shown to have exceptional toughness,

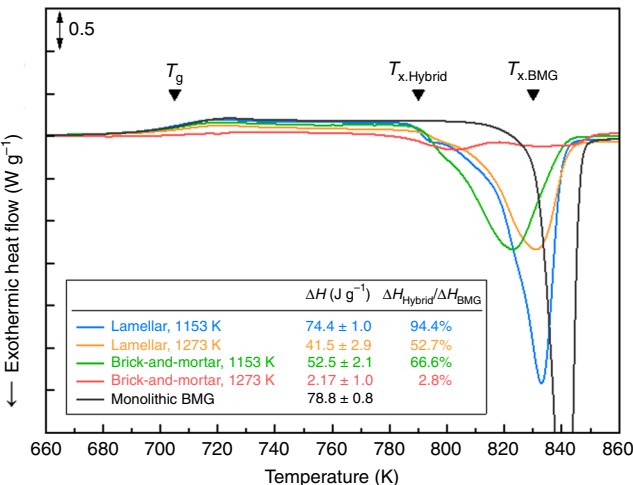

**Fig. 5** Differential scanning calorimetry results for the compliant-phase ceramics. DSC traces for the alumina/BMG-forming materials with lamellar and brick-and-mortar structures infiltrated at 1153 and 1273 K. $T_g$ is the glass-transition temperature, $T_x$ is the onset temperature of crystallization, and $\Delta H$ is the heat of crystallization. The amorphous fraction in the metallic phase is determined by comparing the change of enthalpy during this crystallization event in the infiltrated ceramic–glass samples to the change of enthalpy for the monolithic BMG

and theoretical modeling has predicted that the mechanical properties may be even better with a metallic mortar[8]. To overcome the issues associated with infiltrating a metal into a ceramic scaffold, we have employed the concept of reactive wetting by utilizing a Zr-based BMG as the infiltrating phase. We discuss below the features of our processing procedures to create these materials, our associated attempts at tailoring the ceramic–metal interfaces, and finally how these factors can result in enhanced damage-tolerant properties in nacre-like structural ceramics.

The excellent wettability allowed spontaneous infiltration of the nacre-like alumina scaffolds at very rapid rates (10 min hold time) without the need for applied pressure. This was possible due to the high bonding strength between the alumina and the BMG melt, which can be up to an order of magnitude higher than the nonreactive metal/alumina couples. This was measured using the work of adhesion, which was calculated from surface tension and the final contact angles, as shown in Supplementary Note 1 of the Supplementary Information. This high bonding strength also leads to high capillary pressures induced by the BMG melt. Based on the calculations in Supplementary Note 2, we found that the capillary pressures induced by the BMG melt are the same or higher than the external pressure applied for complete infiltration in other experiments, leading to spontaneous infiltration. This is true despite the high viscosity of the BMG melt because the length scale of the ceramic scaffold is sufficiently large to accommodate the infiltration kinetics of the system according to Darcy's law, as shown in Supplementary Note 3. However, infiltrating finer-scale scaffolds (similar to the dimensions found in natural nacre) with this BMG would present a problem as the high viscosity of the BMG melt would prevent complete infiltration (Supplementary Fig. 2). This indicates that for future synthesis of nacre-like materials using infiltration, it is critical to observe the length scale of the scaffolds along with the viscosity and wettability of the melt to find feasible processing conditions.

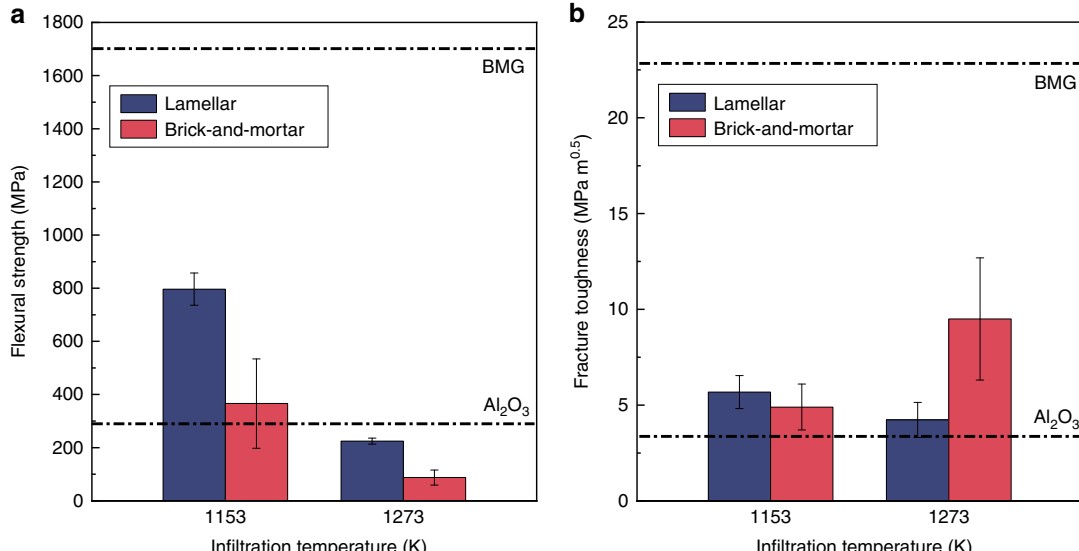

**Fig. 6** Flexural strength and fracture toughness for compliant-phase ceramics. **a** Influence of infiltration temperature on the flexural strength of the lamellar and brick-and-mortar samples. The strength of the samples can reach as high as 800 MPa for lamellar samples and 400 MPa for brick-and-mortar samples. While the flexural strength is dwarfed by its BMG constituent, its strength is significantly higher than in pure alumina, the main component of the brick-and-mortar materials. This flexural strength decreases by 75% when infiltrated at higher temperatures. **b** The fracture toughness of the brick-and-mortar materials is higher than for monolithic alumina, the main component of our compliant-phase ceramics. The marked effect of the microstructure is evident by the fact that the lamellar structures show very little change in toughness when infiltrated at different temperatures, whereas the brick-and-mortar samples show a large increase in fracture toughness from about 5 to 8 MPa·m½ (with the R-curve rising as high as 14 MPa·m½ (Fig. 7)), when infiltrated at higher temperatures

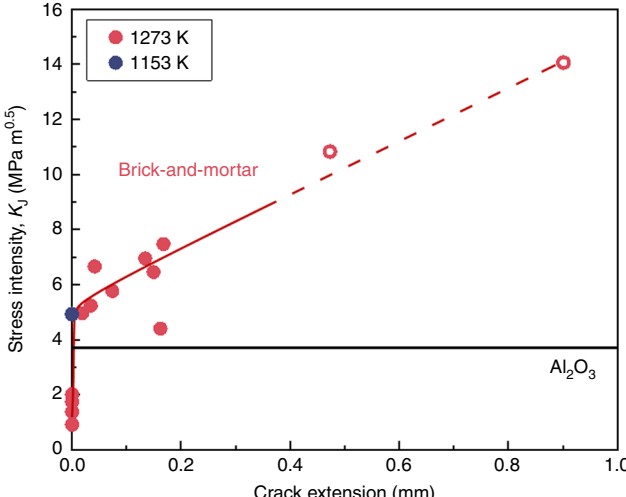

**Fig. 7** Crack-resistance curves (R-curves) for brick-and-mortar alumina/BMG-forming materials infiltrated at 1273 K. The onset of crack initiation and immediate catastrophic fracture of samples infiltrated at 1153 K is included for reference. Comparison of the fracture toughness between the samples infiltrated at the two infiltration temperatures indicates how an appropriately created brick-and-mortar microstructure can serve to stabilize the slow growth of incipient cracks, manifested in the form of a rising R-curve, rather than leading to unstable catastrophic fracture at the onset of crack initiation. (The open circles refer to data points that are not strictly valid according to ASTM Standard E1820[49], as they exceed the maximum crack extension capacity for the test specimens that were used)

The temperature used for infiltration had a marked effect on the flexural strength and fracture toughness of both the lamellar and brick-and-mortar structures. This is particularly clear for the samples with a nacre-like architecture, where samples infiltrated at 1273 K compared with the materials infiltrated at 1153 K displayed a factor-of-four lower strength, yet a factor-of-three higher toughness, caused by stable crack growth (and rising R-curve behavior) rather than sudden catastrophic fracture. The latter characteristic of sustaining stable cracking following crack initiation, without immediate catastrophic failure, represents an essential property of structural materials. For the case of the 1273 K infiltrated brick-and-mortar materials, this resulted from significant crack propagation along the BMG-mortar/alumina–ceramic interfaces. Indeed, all these strength and toughness properties are primarily related to the nature of these interfaces.

In metal–ceramic systems showing reactive wetting, the interfacial bonding is affected by the chemical reaction between the melt and ceramic[50]. Furthermore, as noted above, the BMG melt infiltrated into the scaffolds can be partially or fully crystallized during solidification due to the interface that acts as a site for heterogeneous nucleation. Thus, the effect of temperature on the reaction and crystallization of the BMG mortar are vital issues to be considered to evaluate the interfacial bonding in these nacre-like alumina/glass-forming hybrid materials. Figure 10a, b shows the interface of the hybrids with a lamellar structure that were infiltrated at 1153 and 1273 K. The material infiltrated at 1153 K (Fig. 10a) shows sharp interfaces between the metallic mortar and alumina, and a faceted crystalline ($Zr_2Cu$) phase with a thickness of $0.3 \pm 0.2$ μm. Reaction products like $ZrO_2$ were not observed at the resolution of the SEM, indicating the slow kinetics of the reaction ($3Zr + 2Al_2O_3 \rightarrow 4Al + 3ZrO_2$) at 1153 K. The material infiltrated at 1273 K (Fig. 10b) shows erosion of the interface, which produces a rough boundary, due to the reaction between the BMG melt and alumina. It should be noted that voids

(marked by yellow arrows) and cracks are concentrated in the layer with a thickness of less than 2 μm between the faceted $Zr_2Cu$ phase and alumina (marked as A in Fig. 10b). The layer also contained round-shaped pores (marked by blue arrows) that were generated from grain pullout during mechanical polishing. Examining this morphology, it is clear that the interfacial intermetallic layer provides a weak interface that creates a tortuous path for crack propagation. This is further supported by the micro-cantilever experiments, which showed a clearly weaker interface strength for the materials infiltrated at 1273 K (Fig. 9d, f), as compared with those infiltrated at 1153 K (Fig. 9c, e).

During solidification, the undercooled BMG melt experiences abrupt volume contraction accompanied by crystallization. The Zr-based BMG used in this study shows about 1.5% volumetric contraction if the BMG melt is fully crystallized at its melting point[51]. To estimate the degree of volume contraction in the metallic mortar, the crystallinity of the metallic mortar was evaluated using a comparison of the heat of crystallization from the alumina/glass materials with the heat of crystallization from the monolithic BMG from the DSC analysis[52] (Fig. 5). The alumina/glass-forming alloy materials with a lamellar structure infiltrated at 1153 K and 1273 K had a crystallinity of 6% and 47%, respectively, which is comparable with the ratio between the thickness of the crystalline phase and that of metallic mortar in the materials with a lamellar structure (5% and 48%, respectively), as confirmed by microstructural analysis. Thus, a volume contraction of more than 0.7% takes place in the metallic mortar when the lamellar scaffold is infiltrated at 1273 K and then water-quenched. Furthermore, the alumina/glass-forming alloy materials with the brick-and-mortar structure infiltrated at 1273 K experience a volume contraction of the metallic mortar of about 1.5% due to their high crystallinity (97%). The significant volume contraction within this material leads to the formation of interfacial cracks or voids, resulting in a low flexural strength in the nacre-like materials.

As the brittle interface forms at higher temperatures, the crack patterns are more tortuous, making it harder to propagate through the sample. The fracture toughness tests further support this hypothesis because the brick-and-mortar samples infiltrated at 1273 K show significant crack deflection compared with those infiltrated at 1153 K, as shown in Fig. 7. The brick-and-mortar material infiltrated at 1153 K has a much higher flexural strength associated with its stronger ceramic/metal interface, but it appears to be too strong as the ceramic bricks fracture. In contrast, the brick-and-mortar samples infiltrated at 1273 K show several extrinsic toughening mechanisms (typical of natural nacre[1,4]), such as inter-brick displacements, brick pullout, and crack deflection, leading to a significant increase in fracture toughness, compared with the corresponding structures infiltrated at 1153 K. However, the decreased flexural strength of samples infiltrated at 1273 K compared with those infiltrated at 1153 K is a strong indication that the alumina/BMG interfacial strength decreases with higher infiltration temperatures.

So, what do we learn from this work with respect to ceramic brick-and-mortar structures? First, we have shown that it is possible to rapidly infiltrate a ceramic scaffold with a metal mortar without any applied pressure using reactive wetting. Moreover, this can result in a high-strength structure due to a strong ceramic/metal (brick/mortar) interface in a high-volume-fraction ceramic structure, as shown for infiltration at 1153 K. However, for toughness, we also require some degree of ductility or shielding, ideally with a structure comprising a high-aspect ratio, fine scale, and bricks in conjunction with a mortar that permits some degree of inter-brick displacements to alleviate any locally high stresses. We achieved this with the structure infiltrated at 1273 K. Instead of outright catastrophic failure, the

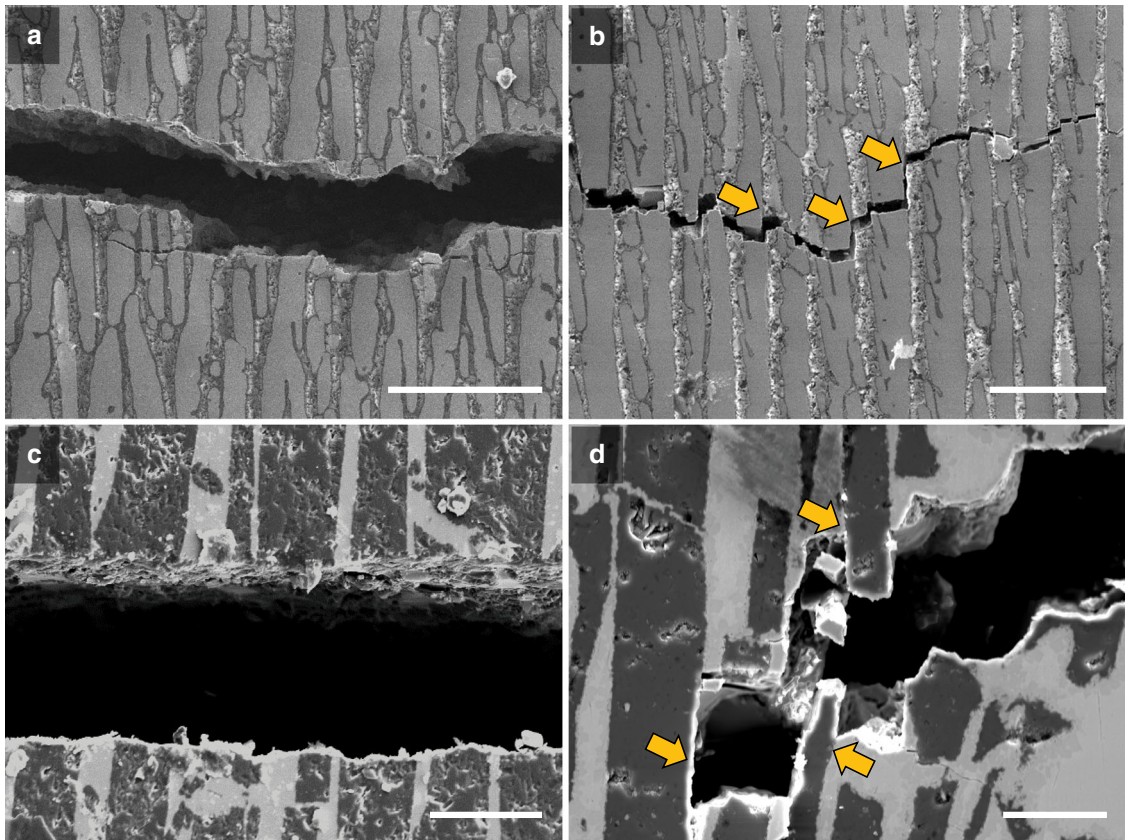

**Fig. 8** SEM micrographs of crack patterns of the infiltrated alumina/BMG-forming materials samples. **a–d** Lamellar (**a**, **b**) and brick-and-mortar (**c**, **d**) samples infiltrated at 1153 K show little to no crack deflection behavior, in contrast to the corresponding samples infiltrated at 1273 K which exhibit crack deflection due to crack propagation along the interface of the ceramic and metallic components, which in turn leads to ceramic brick pullout and hence extrinsic toughening from crack bridging. The arrows highlight where the cracks propagated along the interface of the ceramic and metallic phases. This behavior is critical for stable crack-growth behavior and increases the crack-growth fracture toughness in the brick-and-mortar samples primarily due to crack bridging; however, superior damage tolerance would be generated if the inter-ceramic brick displacements occurred within the metallic mortar, rather than involving ceramic/metal interface failure. Scale bars: 100 μm (**a**, **b**), 50 μm (**c**), and 30 μm (**d**)

1273 K infiltrated structures displayed markedly rising R-curve behavior (Fig. 7), where the inter-brick displacements serve to stabilize the growth of incipient cracks, over crack extensions up to 1 mm, with crack-growth toughness rising to the order of 14 MPa·m$^{1/2}$. This is what nacre-like structures can offer. The limited inter-brick displacements can lead to crack deflection and most importantly brick pullout and crack bridging, which act to stabilize crack growth; there is little change in crack-initiation toughness as the extrinsic toughening mechanisms solely enhance the crack-growth toughness[6]. This is also underlined by the microfracture experiments (Fig. 9), which clearly show that the interface strength is rather poor and most toughening arises from the microstructural alignment with respect to the crack path. A significant increase in initiation toughness is conceivable, but must involve strengthening of the interface properties.

However, the Achilles' heel of these structures is that the inter-brick displacements need to be within the mortar and not along the interface, as in the present high-toughness structure infiltrated at 1273 K. There are two problems with interface cracking. Whereas it still provides a means to alleviate high local stresses and confer good toughness (which is the basic concept underlying the toughness of nacre), weak interfaces invariably result in low strength, which one can readily appreciate by comparing the flexural strength of the 1273 K infiltrated structure to that infiltrated at 1153 K. Additionally, as noted above, the properties of metal-infiltrated nacre-like structures are predicted to be far superior to those of polymer-infiltrated structures because of the

higher tensile/shear strengths of metal mortars;[10] however, for this to be realized, the inter-brick displacements naturally have to be within the mortar. For brick-and-mortar architectures, such behavior has been achieved in the high-toughness alumina–PMMA structures, where the ceramic–polymer interfaces were strengthened by grafting techniques[11], and in coextruded (but very coarse-grained) alumina–nickel structures[24], but to our knowledge, it has yet to be achieved with nacre-like, fine-grained, metal-infiltrated ceramic structures. One possible method to attain such materials is to select a BMG with higher ductility, such as a Pt-based BMG[53], and freeze-cast a substrate the metal can readily wet to overcome the strength and toughness trade-off.

Concerning the damage-tolerant behavior of nacre-like (brick-and-mortar) alumina ceramics infiltrated with metallic mortar of a Zr-based BMG, we therefore draw the following conclusions.

Alumina/BMG-forming hybrid materials, with nacre-like structures containing a high alumina fraction up to 80 vol.%, were rapidly synthesized using freeze-casting followed by melt-infiltration. The excellent infiltration behavior, showing rapid and complete infiltration without any applied pressure, was attributed to the strong chemical interaction and excellent wetting behavior between the BMG melt and alumina. These results show that BMGs, or any alloys with fast and excellent wettability on ceramic, can be infiltrated in a high-volume-fraction ceramic preform with a complex architecture to synthesize advanced bioinspired structural materials with exceptional mechanical properties.

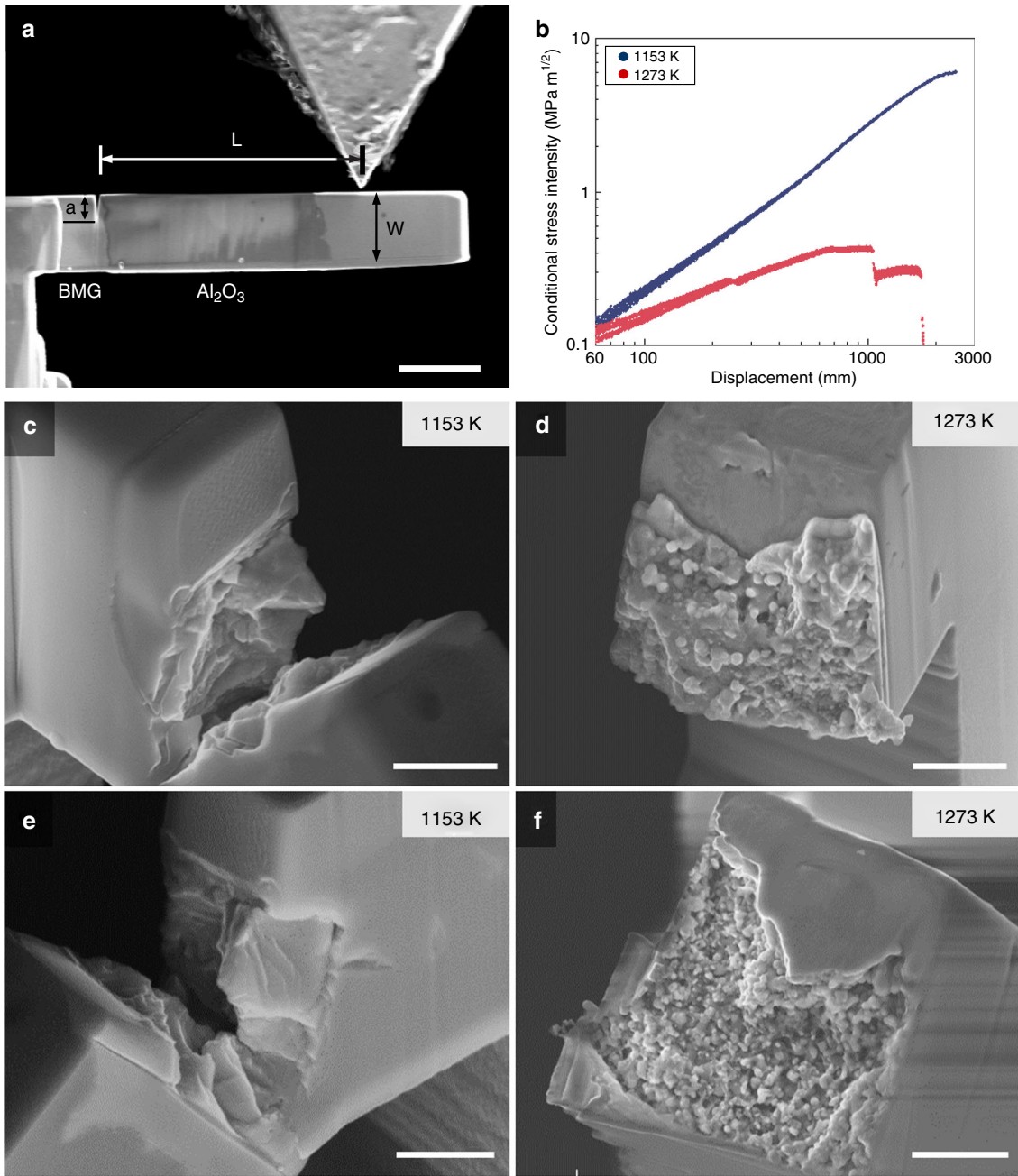

**Fig. 9** SEM micrographs and results of micro-cantilever tests. **a** Representative micro-cantilever specimen for interface toughness measurement. **b** Conditional stress intensity–displacement curves for 1153 K (blue) and 1273 K (red) infiltrated specimens, respectively, on a log–log scale. **c–f** SEM micrographs of the fracture surfaces of the same 1153 K (**c**, **e**) and 1273 K (**d**, **f**) infiltrated specimens, showing a clear distinction between interfacial fracture (**d**, **f**) and fracture through the BMG phase (**c**, **e**). Scale bars: 5 µm (**a**), 1 µm (**c–f**)

Mechanical testing results indicated how infiltration temperature can have a marked effect on the interfacial properties of the resulting hybrid materials. The reactive wetting of BMG with alumina provided a strong interfacial bonding when infiltrated at a lower temperature (1153 K), but the chemical reaction at a higher temperature (1273 K) and crystallization of the BMG melt resulted in the formation of weak ceramic–metal interfaces. These effects of infiltration temperature resulted in a trade-off between the strength and toughness of the compliant-phase ceramic materials. In samples infiltrated at 1153 K, the high interfacial strength led to higher strength in the alumina/BMG-forming hybrids, but low toughness due to the brittle nature of the metallic matrix. The samples infiltrated at 1273 K exhibited high fracture toughness, with stable crack growth, and hence resistance-curve behavior, up to stress intensities of 9–14 MPa·m½ (the latter value being not strictly valid by ASTM Standards[49]), due to interphase displacements between the bricks. However, their strength was lower because these displacements (and ultimately failure) occurred along the interface, as opposed to within the mortar layer, such that nacre-like failure mechanisms were not perfectly mimicked.

## Methods

**Materials processing**. To examine the conditions for metal infiltration into the freeze-cast alumina scaffolds, both lamellar and the final brick-and-mortar structures were processed. These ceramic scaffolds were first fabricated by freeze-casting

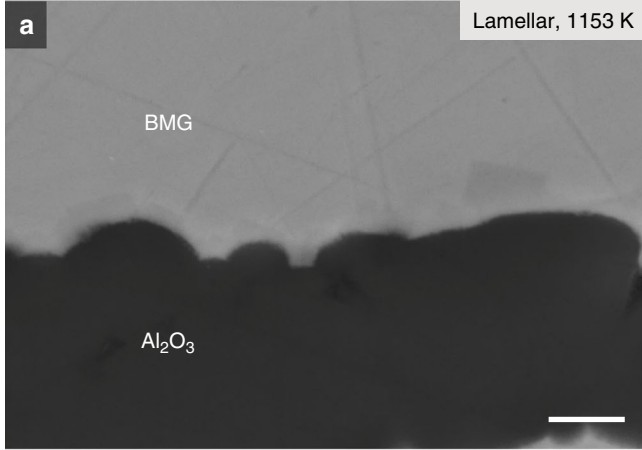

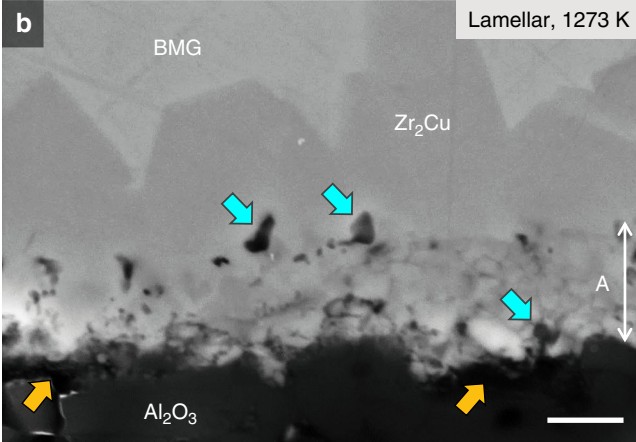

**Fig. 10** SEM micrographs showing the interface of the compliant-phase alumina/BMG-forming materials with a lamellar structure infiltrated at 1153 and 1273 K. **a** The material infiltrated at 1153 K displays a sharp interface between the metallic mortar and alumina, and faceted crystalline ($Zr_2Cu$) phase with a thickness of $0.3 \pm 0.2 \, \mu m$. **b** In contrast, the interface of the material infiltrated at 1273 K reveals erosion of the interface, which produces a rough boundary, due to the significant reaction between the BMG melt and alumina. Scale bar: 1 μm

a water-based suspension consisting of 50 wt.% alumina powder (nanocrystalline, average particle size 150 nm α-$Al_2O_3$ powder, Inframat Advanced Materials, Manchester, CT). The suspensions were formulated with 2.5 wt.% powder of polyacrylic acid (PAA) (98–99% hydrolyzed, high molecular weight, Alfa Aesar, Tewksbury, MA) which was added as a dispersant. Hydrochloric acid (25% (v/v) aqueous solution, Ricca Chemical Company, Arlington, TX) was also used to maintain a pH at about 2 to prevent agglomeration. The samples were milled to mix and homogenize for a minimum of 12 h.

The samples were placed in a cylindrical Teflon mold with a copper cold finger at the bottom. The suspension in the mold was subjected to controlled directional freezing, which allows lamellar scaffolds to form. The cold finger was cooled to 113 K at a rate of −5 K/min, with its temperature controlled with a liquid-nitrogen cold bath and ring heater. To prepare samples with a lamellar structure aligned over several centimeters, the materials were first cast using a polydimethylsiloxane 20° angle wedge to induce bidirectional freezing, as discussed elsewhere[18]. After complete freezing, the samples were removed from the mold and placed in a freeze drier (Freeze Dryer 8, Labconco, Kansas City, MI). By holding the samples for 3 days in a low-pressure (0.06 mbar), ambient-temperature environment, the ice was sublimated to form green ceramic lamellar scaffolds, which were subsequently fired for 2 h in air at 873 K to eliminate the organic additives before being sintered at 1823 K for 3 h (Air furnace: 1216BL, CM Furnaces Inc., Bloomfield, NJ) to densify the ceramic lamellae. To convert these lamellar structures into a brick-and-mortar form, they were infiltrated with paraffin wax and then compacted in a hydraulic press at a pressure of 1200–1400 MPa to ensure a high ceramic content, with plates heated at 353 K to allow the wax to flow. After pressing, the samples were fired and sintered using the same procedure described above and subsequently cut into $2.5 \times 2.5 \times 12$–15 mm pieces before infiltration with the BMG.

The $Zr_{46}Cu_{30.14}Ag_{8.36}Al_8Be_{7.5}$ (at.%) BMG ingots, which have a high glass-forming ability that can be cast up to a critical diameter of 73 mm[54], were fabricated by arc melting a mixture of their constituent elements (Ag, Al, Cu, and Zr with a purity of above 99.9%) and a commercial Cu–Be alloy ($Cu_{77.3}Be_{22.7}$) under a Ti-gettered Ar atmosphere. The ingots were re-melted three times to ensure compositional homogeneity before the alloys were cast into a copper mold (4 mm in diameter, 50 mm in length).

The alumina/BMG-compliant-phase ceramic was prepared by pressureless melt-infiltration into the alumina scaffold. Specifically, the scaffold was placed in a quartz tube with an inner diameter of 5 mm, and then the tube was necked 3 mm above the scaffold to prevent floatation during the infiltration. The as-cast BMG rod was placed on the neck, and then the tube was sealed in vacuo using a rotary pump to prevent the oxidation of the as-cast rod during the processing. The sealed tube was placed in an electric resistance furnace heated to 1153 or 1273 K, i.e., above the melting temperature of the BMG (1103 K). After holding for 3 min, the tube, where the molten BMG was trapped on the neck due to its high viscosity and surface oxide layer, was tapped on the bottom of the furnace to dispense fresh BMG melt under the neck. The scaffold immersed in the BMG melt was held for 10 min during which the scaffold was infiltrated without external pressure. The tube after infiltration was removed from the furnace and quenched in water.

**Measurement of thermophysical and wetting properties**. Thermophysical properties of the BMG were determined by an electrostatic levitation method (ESL) in Japan Aerospace Exploration Agency (JAXA). Ball-shaped samples of ~2 mm in diameter were prepared by arc melting pieces of the as-cast BMG rods. The spherical samples were levitated in a high-vacuum environment (~$10^{-5}$ Pa) using electrostatic forces. Samples were heated and melted by $CO_2$ lasers, and the temperature was measured using pyrometers at 120 Hz acquisition rate. Surface tension and viscosity of the $Zr_{46}Cu_{30.14}Ag_{8.36}Al_8Be_{7.5}$ alloy were determined using the oscillation-drop method, as described elsewhere[55].

Contact angles of the BMG on the polycrystalline alumina substrate were measured in a vacuum of ~$10^{-4}$ Pa by the sessile-drop method, utilizing a stainless-steel vacuum chamber equipped with a drop dispenser and a tube furnace[56]. About 100 mg of the BMG sample was prepared by arc melting pieces of the as-cast BMG rods. The alumina substrates were prepared by polishing with diamond pastes to give an average surface roughness below 100 nm. The substrates were positioned ~5 mm below the dispenser, and the BMG sample was heated in the drop dispenser made of an alumina ceramic tube with a small 1 mm hole at the tip. The BMG melt was pushed through the hole by applying argon pressure so that a potential oxide film on the melt can be fractured and a fresh droplet can be supplied on the substrate. The experiment was observed through a hole in the furnace, and images of the droplet were recorded at a speed of 10 frames per second by using a high-speed digital camera.

Phase constitution was confirmed by X-ray diffraction (XRD; New D8 Advance, Bruker Corporation, Karlsruhe, Germany) using monochromatic Cu Kα radiation operated at 40 kV and 40 mA. Thermal analysis of the alumina/BMG ceramic was conducted by differential scanning calorimetry (DSC, DSC 8500, Perkin Elmer, Waltham, USA) using a constant heating rate of 40 K/min. The microstructures of the $Al_2O_3$/BMG ceramic were analyzed using scanning electron microscopy (SEM; SU70, Hitachi Corp., Tokyo, Japan) equipped with energy-dispersive X-ray spectroscopy.

**Materials and microscale interface characterization**. After casting, the samples were cut and polished for mechanical testing and characterization. Both ends of the ingot were machined off using a water-cooled low-speed circular saw to reveal the cross-sectional region of the material, which was then ground into rectangular beams (2.3–2.5 mm wide, 2.3–2.5 mm thick, and 12–15 mm in length) using SiC papers with grits ranging from 60 to 1200, with a final polish with a 1 μm diamond suspension.

To determine the interface fracture strength between the ceramic bricks and the BMG mortar, samples from both infiltration temperatures were tested using notched micro-cantilever beams that were focused ion beam milled (FIB, LEO 1540 XB, Carl Zeiss AG, Oberkochen, Germany) at 30-kV acceleration voltage with ion currents subsequently decreasing from 10 nA to 500 pA. The initial notch with a radius of ~20 nm was placed at the interface by utilizing a current of 100 pA. All micro-cantilever fracture experiments were subsequently conducted in situ in an SEM (DSM 982, Carl Zeiss AG, Oberkochen, Germany) using a Hysitron picoindenter PI-85 (Bruker Corporation, Billerica, USA) equipped with a wedge-shaped conductive diamond tip (Synton-MDP AG, Nidau, Switzerland), operated in open-loop mode with a loading rate of 10 μN/s. To compare the interface fracture characteristics, the conditional stress intensity vs. displacement was calculated from the load–displacement data as outlined by Wurster et al.[57] with the geometric quantities indicated in Fig. 9a.

**Macroscale strength and toughness measurements**. The macroscopic flexural strength of the samples was measured using a minimum of three samples for each composite that were tested using three-point bend testing on unnotched specimens, with a loading support span of 10 mm, at a displacement rate of 1 μm/s, in general

accordance with the ASTM Standard D790[58]. These tests were performed on an Instron 5944 electromechanical testing system (Instron Corporation, Norwood, MA, USA).

Fracture toughness measurements were conducted on single edge-notched bend (SEN(B)) samples, which were notched using a low-speed diamond saw. The notch root was then sharpened using a micro-notching technique which involves polishing the end of the notch with a razor blade under load, and immersed in a 6 μm diamond slurry; the resulting notch root radii were typically on the order of 10 μm. These samples were loaded at a displacement rate of 0.55 μm/s in three-point bending, with a loading span of 10 mm, in general accordance with ASTM Standard E1820[49]. A minimum of three samples were tested for each composite. Tests were performed using a Deben MicroTest 2kN (Deben, UK) bending stage embedded in a Hitachi S-4300SE/N (Hitachi America, Pleasanton, CA, USA) SEM to permit real-time observations of the crack path and its interaction with the prevailing microstructure during the measurement of the fracture toughness and crack-resistance curves.

Due to the relatively small size of the fracture toughness samples and the fact that these compliant-phase ceramic materials display some degree of inelasticity, nonlinear–elastic fracture mechanics measurements were made to determine $J$-based crack-resistance curves ($J$-$R$ curves), where $J$ is the $J$ integral, i.e., the field parameter that characterizes the local nonlinear–elastic (Hutchinson–Rice–Rosengren, HRR) stress and displacements fields in the immediate vicinity of a crack tip in a nonlinear–elastic solid. The R-curve assesses the value of $J$ required to sustain stable crack extension. In the current tests, such crack extension was measured using the elastic unloading compliance of the specimen, whereas the corresponding $J$ values were computed in terms of their elastic and plastic components ($J = J_{el} + J_{pl}$). The elastic component of $J$ was calculated from the linear–elastic stress-intensity value, $K$, i.e., $J_{el} = K^2/E'$, where $E' = E$ (Young's modulus) in plane stress and $E/(1-\nu)^2$ in plane strain ($\nu$ is Poisson's ratio). The elastic modulus and Poisson's ratio were 238.9 GPa and 0.3, respectively; the modulus was calculated using the rule of mixtures based on the elastic modulus of the monolithic samples. The plastic component was determined in terms of the plastic portion of the area under the load–displacement curve, $A_{pl}$, using the expression $J_{pl} = 1.9A_{pl}/Bb$, where $b$ is the uncracked ligament size and $B$ is the sample thickness[49]. Equivalent stress-intensity values were then back-calculated from these measured $J$ values using the standard mode I $J$-$K$ equivalence, i.e., $K_J = (J E')^{1/2}$.

## Data availability

The data that support the findings of this study are available from the corresponding authors, at espark@snu.ac.kr or roritchie@lbl.gov, upon reasonable request.

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

## Acknowledgements

This work was supported by the Mechanical Behavior of Materials Program (KC-13) at the Lawrence Berkeley National Laboratory (LBNL), funded by the U.S. Department of Energy, Office of Science, Office of Basic Energy Sciences, and Materials Sciences and Engineering Division, under contract no. DE-AC02-05CH11231. A.W. acknowledges support by an individual National Science Foundation Graduate Research Fellowship (Grant no. DGE 1106400). The authors would like to thank James Wu for use of his processing facilities at LBNL, Geun Woo Lee for helpful discussions, and Valentina Naglieri and Shinhoo Woo for their processing assistance. E.S.P. was supported by the National Research Foundation of Korea grant funded by the Korean government (Ministry of Science and ICT) (NRF-2017R1A2B2007874) and the Institute of Engineering Research at Seoul National University, Korea. D.K. acknowledges financial support from the European Research Council under Grant number 771146 (TOUGHIT). This is a U.S. Government work and not under copyright protection in the U.S.; foreign copyright protection may apply (2019).

## Author contributions

R.O.R. and E.S.P. designed the research; A.W. and J.I.L. processed the alumina–glass ceramics, with help from A.P.T.; thermophysical properties were determined by J.I.L., C. W.R., J.Y.K., T.I., J.S., and A.M.; mechanical properties were measured by A.W. and B.G. (macroscale) and M.A. and D.K. (microscale); all authors analyzed and interpreted the data, and A.W., J.I.L., E.S.P., and R.O.R. wrote the paper.

## Additional information

**Competing interests:** The authors declare no competing interests.

