## [Peer Review File · Nature Communications]

Reviewers' comments:

Reviewer #1 (Remarks to the Author):

This manuscript describes the preparation of metal-ceramic composites with a high ceramic volume fraction and a nacre-inspired brick-and-mortar structure via simple freeze-casting and pressureless-infiltration techniques based on the good wettability between the Zr-based alloy with excellent BMG-forming ability and Al₂O₃ ceramic. The article is well written with rich content, clear logic, and reasonable analysis. Although the preparation of the metal-ceramic composites with the nacre-like structures has recently widely reported in literature, the work described in this article is still innovative and instructive for researchers engaged in the related field. Unfortunately, due to the inherent low plasticity of the amorphous alloy, the resultant composite fails to achieve perfect combination of strength and toughness as the natural material they desired to imitate.

Herein, I have some questions for the authors to consider:

(1) The title of this article is not rigorous enough. Although the authors used the Zr-based bulk amorphous alloy, the primary phases in the resultant composites are not amorphous, especially in those with brick-and-mortar structure; i.e., the original amorphous alloy is basically crystallized (Figure 3b). Therefore, it is better to use "bulk-metallic glass-forming alloy" rather than "bulk-metallic glass ceramics" in the title. Also, why did they use "ceramics" not "composites"?

(2) In the past few years, a lot of work has been carried out on the preparation of the metal-ceramic composites with nacre-like (mainly lamellar) structures by using the similar techniques (freeze casting combined with pressure or pressureless infiltration). However, the authors did not fully review and comment on these work in the introduction.

(3) From the inset in Figure 2, it can be seen that the authors used a dispensed sessile method to measure the wettability between the Zr-based BMG-forming alloy and alumina. Considering that the Zr-based amorphous alloy is wettable with most ceramics (e.g., alumina), and reactive with graphite (which is usually used as crucible for melting the alloy), a key question is: what material did the authors use as dispenser in the wetting experiments (for pre-melting the amorphous alloy). The process for the wetting test needs to be described in more detail.

(4) It seems more important for the authors to test the wettability of the porous freeze-cast alumina scaffolds (after sintering and after compaction) by the Zr-based BMG-forming alloy and the corresponding infiltration kinetics. Due to the particularity of the surface structure of the freeze-cast samples, their wettability could be very different from that of the dense alumina substrates.

(5) In the experimental part, the authors indicate that the volume fraction of Al₂O₃ in the alumina slurry is 37%. What is the real density in the body after sintering at 1550°C for 3h? (In page 6, the authors gave a value of 37 vol.% based on image analysis. In my opinion, volume shrinkage should take place during the high-temperature sintering) In addition, the authors should clearly indicate the particle size of the alumina powders they used (rather than a simple indication of sub-micrometer), since the particle size affects the freezing behavior and the final microstructure.

(6) The authors should give the stress-strain curves or load-displacement curves obtained in the bending strength and fracture toughness tests in their supplementary materials and indicate the number of the samples they used to test the mechanical properties. Also, they should explain how the physical parameters, such as elastic modulus E and Poisson's ratio ν which are necessary for calculating the fracture toughness, were determined.

(7) In view of the mechanical performance, for the lamellar composites, the change in bending strength with infiltration temperature is large (about 3 times), but the difference in fracture toughness is small. However, for the brick-and-mortar samples, the bending strength is very low (about 1/10 of the lamellar composite), but the fracture toughness is really high (about 3 times that of the lamellar composite). This is somewhat confusing even though the authors gave explanations from the interfacial structure analysis. It is necessary for the authors to provide the original stress-strain or displacement-load curves obtained in the mechanical property tests instead of the processed data (Figure 7), so that the readers can better understand their results.

(8) As for Figure 7, the authors believe that a higher toughness number is not strictly valid in terms of ASTM Standard E1820 when the stress intensity exceeds 9. However, in their abstract

and conclusions, they still used the highest toughness number (14 MPa.m^{1/2}) to characterize their material. This is obviously not very strict.

Reviewer #2 (Remarks to the Author):

Biogenic nacre is exceptional for its outstanding mechanical properties although the properties of the single components are relatively meager. The outstanding properties arise from well-optimized and hierarchically structured design by which the interplay between inorganic majority and organic minority components can dominate the material's response to external load and fracture. It is an intrinsic driving force in the development of biomimetic materials engineering is the effective adaptation of these evolutionary designs and their further enhancement to make them applicable for a wider field of materials and environments. A cavalcade of papers has been published on how to understand, faithfully mimic and exploit this finely-tuned biocomposite. One of these papers showed by simulations that by replacing the soft organic matrix by a metal can further enhance the performance of nacre-like material and, naturally, would allow usage of such metal/ceramic composites in settings which would inevitably destroy the organic minority phase. The manuscript of Ritchie et al. addresses this very question and reports on the attempt to develop a nacre-like alumina/metal composite ceramic. From an engineering perspective, this contribution is very revealing concerning how to process and successfully achieve metal infiltration of ceramic bodies. Further, the mechanical performance which the nacre-like composite achieves, is clearly impressive. However, the paper also shows a distinct difference to biogenic nacre as its mechanical properties are dominated by the interfacial failure at the composite interface. In biogenic nacre, this is one of the key aspects by which nacre achieves its near-to unparalleled characteristics. Overall, I see this paper as an important step towards transgressing the compositional limits of biomimetic ceramic composites. The scientific work was carried out and is presented in a clear and stringent way. As a matter of fact, I have no major concerns arising from the article itself – minor ones are listed below. With that I congratulate the authors to the excellent work.

My only major concern is that I doubt that Nature Communications might be the suitable journal for this contribution. I see Nature Communications as a multidisciplinary journal, covering the whole range of the natural sciences, and contributions published by Nat Commun should also be of interest for neighboring fields. In its current state, this manuscript is motivated to answer a major question for the field of bioinspired materials science but, in the end, it cannot provide a satisfying answer. The results underline the issue of interfacial bonding in such metal/ceramic composites and that future research have to explicitly address this. In a nutshell, the manuscript is about successful metal infiltration of a ceramic scaffold—the latter is generated by methods already long-established (freeze-casting)—and the still prevailing issue of metal-ceramic bounding. With that, it may not be attractive for the broad readership of Nat Commun. A journal which is more dedicated to such material engineering questions, e.g., Adv. Eng Mater.

Minor comments

- The nacre structure is an abundant one in calcareous organisms, not only abalone generates this laminar structure but also many other gastropods and also in an overwhelming number of bivalves.
- Mineral bridges are important for the mechanical properties, but this feature is not present in all nacre subtypes; the nacreous of some calcifying organisms have no bridges but rely on surface asperities.
- Mortar is maybe not a good term for the compliant biopolymer phase. Also, for readers from a non-MSE background, BMGs should be introduced as amorphous alloys in the first instance, to prevent misunderstanding/incomprehension.
- A short(!) overview of other successful generation of nacreous-like structures should be provided; explicitly for the successful exploitation of freeze-casting in these approaches-

Reviewer #3 (Remarks to the Author):

This is a wonderful paper describing the way to bio-inspired new type of brick-and-mortar based ceramics using glass-ceramics in addition. The work is new in its detailed experimental and numerical Analysis of these materials. It thus will inspire the community to additional and further new work in the field. The work can be reproduced as it is given in the manuscript and gives detailed support in understanding the way of deriving the results. The report is written in perfect English. It thus deserves to be published as it is.

Response to Reviewers

Reviewer #1 (Remarks to the Author):

This manuscript describes the preparation of metal-ceramic composites with a high ceramic volume fraction and a nacre-inspired brick-and-mortar structure via simple freeze-casting and pressureless-infiltration techniques based on the good wettability between the Zr-based alloy with excellent BMG-forming ability and Al₂O₃ ceramic. The article is well written with rich content, clear logic, and reasonable analysis. Although the preparation of the metal-ceramic composites with the nacre-like structures has recently widely reported in literature, the work described in this article is still innovative and instructive for researchers engaged in the related field. Unfortunately, due to the inherent low plasticity of the amorphous alloy, the resultant composite fails to achieve perfect combination of strength and toughness as the natural material they desired to imitate. Herein, I have some questions for the authors to consider:

(1) The title of this article is not rigorous enough. Although the authors used the Zr-based bulk amorphous alloy, the primary phases in the resultant composites are not amorphous, especially in those with brick-and-mortar structure; i.e., the original amorphous alloy is basically crystallized (Figure 3b). Therefore, it is better to use "bulk-metallic glass-forming alloy" rather than "bulk-metallic glass ceramics" in the title. Also, why did they use "ceramics" not "composites"?

The reviewer's comment is very helpful. We have changed the title of the manuscript according to the suggestion by the reviewer. However, in the manuscript, we described our materials as "ceramics" to recognize that we are trying to simulate nacre with its 95 vol.% ceramic (mineral) content and 5 vol.% bio-mortar. Admittedly, we can only get to just above 80 vol.% ceramic with freeze-casting, but the mortar acts more like a "lubricant" between the ceramic bridging rather than a load-bearing second constituent of a composite. I hope that the reviewer will bear with us, but we feel that the word "composite" conveys the wrong connotation here, and further may draw inappropriate comparisons to "cermets", which were truly ceramic/metal composites (typically with ceramic contents ~50 vol%) that were studied during the 60's. However, the reviewer has a point – these are not monolithic ceramics in the normal sense – and so we have used the term "compliant-phase ceramics" or "hybrids", which are terms that we, as well as others in the biomaterials area, have used frequently in previous papers to describe these type of materials.

(2) In the past few years, a lot of work has been carried out on the preparation of the metal-ceramic composites with nacre-like (mainly lamellar) structures by using the similar techniques (freeze casting combined with pressure or pressureless infiltration). However, the authors did not fully review and comment on these work in the introduction.

The references and introduction have been edited to include further work with pressureless infiltration into freeze-cast scaffolds along with a reference to a review paper with a more comprehensive summary of the current state of freeze-casting bioinspired materials.

(3) From the inset in Figure 2, it can be seen that the authors used a dispensed sessile method to measure the wettability between the Zr-based BMG-forming alloy and alumina. Considering that the Zr-based amorphous alloy is wettable with most ceramics (e.g., alumina), and reactive with graphite (which is usually used as crucible for melting the alloy), a key question is: what material did the authors use as dispenser in the wetting experiments (for pre-melting the amorphous alloy). The process for the wetting test needs to be described in more detail.

We used an alumina tube with a small 1-mm hole at the tip as a dispenser and the Zr-based BMG-forming alloy was heated in the dispenser. Ar gas was applied to the dispenser to supply a drop of the liquid alloy on the alumina substrate. These procedures help to break the oxide film on the liquid alloy which can be formed during heating and supply a fresh droplet on the substrate. More details about the experimental conditions can be found in papers by J. Schmitz et al. [J Mater Sci (2010) 45:2144–2149 (Ref. 48 in the main text)] and [J Mater Sci (2014) 49:2286–2297]. As the reviewer suggested, we revised the description of the sessile-drop method in the Method Section as follows:

Contact angles of the BMG on the polycrystalline alumina substrate were measured in a vacuum of $\sim 10^{-4}$ Pa by the sessile-drop method, utilizing a stainless-steel vacuum chamber equipped with a drop dispenser and a tube-shaped Mo resistance wire furnace.⁴⁸ About 100 mg of the BMG sample was prepared by arc melting pieces of the as-cast BMG rods. The alumina substrates were prepared by polishing with diamond pastes to give an average surface roughness below 100 nm. The substrates were positioned ~ 5 mm below the dispenser, and the BMG sample was heated in the drop dispenser made of an alumina ceramic tube with a small hole of 1 mm at the tip. The BMG melt was pushed through the hole by applying argon pressure so that a potential oxide film on the melt can be fractured and a fresh droplet can be supplied on the substrate. The experiment was observed through a

hole in the furnace, and images of the droplet were recorded at a speed of 10 frames per second by using a high-speed digital camera.

(4) It seems more important for the authors to test the wettability of the porous freeze-cast alumina scaffolds (after sintering and after compaction) by the Zr-based BMG-forming alloy and the corresponding infiltration kinetics. Due to the particularity of the surface structure of the freeze-cast samples, their wettability could be very different from that of the dense alumina substrates.

Reactive wetting in the metallic alloy/ Al_2O_3 system has been studied extensively and is known to exhibit very wide range of contact angles and several other complications (affected by $p(\text{O}_2)$, temperature, crystallographic face of the alumina, among many other factors). In most cases, the contact angle changes significantly due to the reaction at a rate dictated by the reaction kinetics, or by diffusion of reactant in the liquid. This is also affected by the porosity of the substrate. With our Zr-based BMGs near-perfect wetting occurred within seconds for the Zr-based BMG on alumina, consistent with reactive wetting.

Therefore, we recognize how the freeze-cast surfaces may have different wetting angles due to the higher surface roughness than the alumina substrates used in this article. However, the high wettability of BMG-forming alloys on alumina is best represented in a replicable fashion on fully dense alumina substrates. Due to the heterogenous nature of the freeze-cast alumina surface, it is difficult to ensure that we are consistently measuring the wettability of the BMG-forming alloy on alumina because our results may be affected by the various porosities on the surface. This may affect the accuracy and replicability of the sessile drop experiments.

To illustrate this, we also performed a test in order to measure the contact angles of the Zr-based BMG-forming alloy dropped on porous freeze-cast alumina scaffolds with lamella structure. However, as shown in the figures below, the liquid alloy dropped on the surface immediately infiltrated the scaffold within 1 second. This result shows why we measure the wettability using the dense alumina substrate, instead of the freeze-cast alumina scaffolds, and clearly demonstrates excellent wetting behavior of porous freeze-cast alumina scaffolds by the Zr-based BMG-forming alloy melt.

(5) In the experimental part, the authors indicate that the volume fraction of Al_2O_3 in the alumina slurry is 37%. What is the real density in the body after sintering at 1550°C for 3h? (In page 6, the authors gave a value of 37 vol.% based on image analysis. In my opinion, volume shrinkage should take place during the high-temperature sintering) In addition, the authors should clearly indicate the particle size of the alumina powders they used (rather than a simple indication of sub-micrometer), since the particle size affects the freezing behavior and the final microstructure.

The authors of this paper appreciate this particular comment by the reviewer and has edited the methods section of the paper accordingly. The methods now specify that the particle size of the alumina powders is 150 nm and that the suspension is made with 50 wt.% alumina.

(6) The authors should give the stress-strain curves or load-displacement curves obtained in the bending strength and fracture toughness tests in their supplementary materials and indicate the number of the samples they used to test the mechanical properties. Also, they should explain how the physical parameters, such as elastic modulus E and Poisson's ratio ν which are necessary for calculating the fracture toughness, were determined.

We have included the stress-strain curves for the flexural strength tests and included representative load-displacement curves for the fracture toughness tests in the supplementary materials. The methods section has also been edited to indicate how many samples were tested for the mechanical properties of the materials. The elastic modulus and Poisson's ratio were 238.9 GPa and 0.3, respectively; the modulus was calculated using the rule of mixtures based on the elastic modulus of the monolithic samples. This has been added in the methods of the paper. It is important to note that fracture toughness values are the only values that are impacted by the elastic modulus and Poisson's ratio. Any error in these values has very limited impact on the equivalent toughness values.

(7) *In view of the mechanical performance, for the lamellar composites, the change in bending strength with infiltration temperature is large (about 3 times), but the difference in fracture toughness is small. However, for the brick-and-mortar samples, the bending strength is very low (about 1/10 of the lamellar composite), but the fracture toughness is really high (about 3 times that of the lamellar composite). This is somewhat confusing even though the authors gave explanations from the interfacial structure analysis. It is necessary for the authors to provide the original stress-strain or displacement-load curves obtained in the mechanical property tests instead of the processed data (Figure 7), so that the readers can better understand their results.*

The curves have been included in the Supplementary Materials to aid understanding of the results along with a note within the paper to notify readers about the images.

(8) *As for Figure 7, the authors believe that a higher toughness number is not strictly valid in terms of ASTM Standard E1820 when the stress intensity exceeds 9. However, in their abstract and conclusions, they still used the highest toughness number (14 MPa.m^{1/2}) to characterize their material. This is obviously not very strict.*

The authors have edited the abstract and conclusions to state that the stress intensity of the materials are approximately 9 MPa·m^{1/2}, according to the reviewer's comments.

Reviewer #2 (Remarks to the Author):

Biogenic nacre is exceptional for its outstanding mechanical properties although the properties of the single components are relatively meager. The outstanding properties arise from well-optimized and hierarchically structured design by which the interplay between inorganic majority and organic minority components can dominate the material's response to external load and fracture. It is an intrinsic driving force in the development of biomimetic materials engineering is the effective adaption of these evolutionary designs and their further enhancement to make them applicable for a wider field of materials and environments. A cavalcade of papers has been published on how to understand, faithfully mimic and exploit this finely-tuned biocomposite. One of these papers showed by simulations that by replacing the soft organic matrix by a metal can further enhance the performance of nacre-like material and, naturally, would allow usage of such metal/ceramic composites in settings which would inevitably destroy the organic minority phase. The manuscript of Ritchie et al. addresses this very question and reports on the attempt to develop a nacre-like alumina/metal composite ceramic. From an engineering perspective, this contribution is very revealing concerning how to process and successfully achieve

metal infiltration of ceramic bodies. Further, the mechanical performance which the nacre-like composite achieves, is clearly impressive. However, the paper also shows a distinct difference to biogenic nacre as its mechanical properties are dominated by the interfacial failure at the composite interface. In biogenic nacre, this is one of the key aspects by which nacre achieves its near-to unparalleled characteristics.

Overall, I see this paper as an important step towards transgressing the compositional limits of biomimetic ceramic composites. The scientific work was carried out and is presented in a clear and stringent way. As a matter of fact, I have no major concerns arising from the article itself – minor ones are listed below. With that I congratulate the authors to the excellent work.

My only major concern is that I doubt that Nature Communications might be the suitable journal for this contribution. I see Nature Communications as a multidisciplinary journal, covering the whole range of the natural sciences, and contributions published by Nat Commun should also be of interest for neighboring fields. In its current state, this manuscript is motivated to answer a major question for the field of bioinspired materials science but, in the end, it cannot provide a satisfying answer. The results underline the issue of interfacial bonding in such metal/ceramic composites and that future research have to explicitly address this. In a nutshell, the manuscript is about successful metal infiltration of a ceramic scaffold – the latter is generated by methods already long-established (freeze-casting) – and the still prevailing issue of metal-ceramic bounding. With that, it may not be attractive for the broad readership of Nat Commun. A journal which is more dedicated to such material engineering questions, e.g., Adv. Eng. Mater.

Minor comments

- The nacre structure is an abundant one in calcareous organisms, not only abalone generates this laminar structure but also many other gastropods and also in an overwhelming number of bivalves.

The sentence in the introduction has been edited to note how other mollusks generate this microstructure.

- Mineral bridges are important for the mechanical properties, but this feature is not present in all nacre subtypes; the nacreous of some calcifying organisms have no bridges but rely on surface asperities.

The introduction has been edited to emphasize how bridges, like platelets with a dovetail geometry, are found in some species of nacre.

- Mortar is maybe not a good term for the compliant biopolymer phase. Also, for

readers from a non-MSE background, BMGs should be introduced as amorphous alloys in the first instance, to prevent misunderstanding/incomprehension.

The authors appreciate this comment about the need to accommodate non-MSE readers. The introduction to BMGs now indicates its glassy atomic structure.

- A short(!) overview of other successful generation of nacreous-like structures should be provided; explicitly for the successful exploitation of freeze-casting in these approaches-

The introduction has been edited to include a short comment on other methods to create the nacre-like materials and references to other review papers that thoroughly discuss how nacre-like structures have been reproduced using freeze casting.

Reviewer #3 (Remarks to the Author):

This is a wonderful paper describing the way to bio-inspired new type of brick-and-mortar based ceramics using glass-ceramics in addition. The work is new in its detailed experimental and numerical Analysis of these materials. It thus will inspire the community to additional and further new work in the field. The work can be reproduced as it is given in the manuscript and gives detailed support in understanding the way of deriving the results. The report is written in perfect English. It thus deserves to be published as it is.

We thank the reviewer for his/her kind comments.

REVIEWERS' COMMENTS:

Reviewer #1 (Remarks to the Author):

In general, the authors made appropriate revisions. This manuscript can be considered for acceptance after a further minor revision. Two questions: (1) The authors used alumina tube as the dispenser to melt and store the Zr-based BMG-forming alloy. Since the wettability of alumina by the Zr-based alloy is very good, (the alloy can spread on the dense alumina surface or fully infiltrate into the porous alumina body in less than 1 s), then, a problem is: did the Zr-based alloy spread on the (inner) surface of the alumina tube or flow out during heating from its melting point (1103 K) to 1273 K? Is the Al₂O₃ tube suitable for this (dispensed) wetting experiment? (2) Although the Zr-based alloy possesses the excellent wettability with alumina, making the pressureless infiltration feasible, its high strength and low plasticity, to a large extent, limit the achievement of high strength and toughness of the composite with the nacre-like structure. In fact, this alloy cannot play the role of "mortar" in nacre (which acts as a lubricant). I hope that the authors could give some suggestions of material selection in the final discussion based on the results of this excellent work, so that readers can get inspiration from it.

Another minor question: For three-point bending tests, the unnotched specimens were loaded at a displacement rate of 1 mm/sec; while for fracture toughness measurements, the SE(B) samples were loaded at a displacement rate of 0.55 $\mu\text{m}/\text{sec}$ in three-point bending. Why so large a difference in the loading/displacement rate (more than 1000 times!)?

By the way, Figure 3 has some formatting errors that need to be corrected.

Response to Reviewers

Reviewer #1 (Remarks to the Author):

In general, the authors made appropriate revisions. This manuscript can be considered for acceptance after a further minor revision. Two questions:

(1) The authors used alumina tube as the dispenser to melt and store the Zr-based BMG-forming alloy. Since the wettability of alumina by the Zr-based alloy is very good, (the alloy can spread on the dense alumina surface or fully infiltrate into the porous alumina body in less than 1 s), then, a problem is: did the Zr-based alloy spread on the (inner) surface of the alumina tube or flow out during heating from its melting point (1103 K) to 1273 K? Is the Al₂O₃ tube suitable for this (dispensed) wetting experiment?

The Zr-based BMG-forming alloy melt did not flow out of the alumina tube during heating up to 1273 K despite the excellent wettability of alumina by the alloy. In the wetting experiment, a fresh melt droplet was supplied through a small hole (diameter: 1 mm) at the tip of the tube by controlling injection gas pressure.

In detail, the BMG melt was pushed through the hole by applying argon pressure so that any potential oxide film on the melt can be fractured and a fresh droplet can be supplied on the substrate. If the alumina tube is assumed to be a cylindrical tube with a capillary diameter of 1 mm, which is the diameter of the hole, capillary pressure ($\Delta P = \gamma \cos\theta D^{-1}$, where γ , θ , and D are surface tension, contact angle, and capillary diameter, respectively) for the alloy melt in the tube is calculated to be about 4.7 kPa at 1273 K. This calculated pressure is about 60 or 90 times lower than the capillary pressures for the alloy melt in the lamellar scaffolds (270 kPa) or in the brick-and-mortar scaffolds (619 kPa) at 1273 K (details about the calculation are described in Supplementary Note 2). Due to the low capillary pressure in the tube and the high surface tension (1.17 N m⁻¹ at 1273 K, from Figure 1a in the main text) of the Zr-based BMG-forming alloy, the alloy melt can be stored in the tube during heating from its melting point to 1273 K. Therefore, it was suitable to use the alumina tube for the dispensed wetting experiment of Zr-based BMG-forming alloy with optimized injection gas pressure.

(2) Although the Zr-based alloy possesses the excellent wettability with alumina, making the pressureless infiltration feasible, its high strength and low plasticity, to a large extent, limit the achievement of high strength and toughness of the composite with the nacre-like structure. In fact, this alloy cannot play the role of “mortar” in nacre (which acts as a lubricant). I hope that the authors could give some suggestions of material selection in the final discussion based on the results of this excellent work, so that readers can get inspiration from it.

The authors of this paper appreciate this suggestion and have changed the manuscript accordingly. At the end of the discussion, we have included a suggestion to use a ductile mortar and choose an appropriate brick material based on the wettability of the mortar on the substrate.

Another minor question: For three-point bending tests, the un-notched specimens were loaded at a displacement rate of 1 mm/sec; while for fracture toughness measurements, the SE(B) samples were loaded at a displacement rate of 0.55 $\mu\text{m}/\text{sec}$ in three-point bending. Why so large a difference in the loading/displacement rate (more than 1000 times!)?

This was a small typing error that has been fixed. The loading displacement rate of the unnotched specimens is 1 $\mu\text{m}/\text{sec}$, rather than 1 mm/sec. The authors thank the reviewer for his/her attention to detail.

By the way, Figure 3 has some formatting errors that need to be corrected.

Following the reviewer’s suggestion, the authors have carefully corrected the formatting errors in Figure 3b. The authors appreciate this feedback from the reviewer.